# Self-Monitoring Intervention for Adolescents and Adults with Autism: A Research Review

**DOI:** 10.3390/bs13020138

**Published:** 2023-02-07

**Authors:** Yi-Fan Li, Suzanne Byrne, Wei Yan, Kathy B. Ewoldt

**Affiliations:** Department of Interdisciplinary Learning and Teaching, The University of Texas at San Antonio, San Antonio, TX 78249, USA

**Keywords:** self-monitoring, autism, technology, review

## Abstract

The ability to work and function independently is one of the most important skills for the achievement of ideal post-school outcomes. The use of self-monitoring to improve independence and/or reduce undesirable behaviors is an imperative need for individuals with autism. The purpose of this literature review was to examine technology-based self-monitoring interventions for individuals with autism. We used a four-step literature search process to identify studies for review. Online databases, such as ERIC, were used to search for studies. Using four inclusion criteria and PRISMA guidelines for the selection and screening process, we identified 16 studies that met the inclusion criteria. We used coding to summarize the following information from the included studies: participants who met the inclusion criteria, primary dependent variable, primary intervention, and study design. The results of the review revealed three primary functions of technology performed in self-monitoring. The included studies targeted on-task behaviors, skill acquisition, and socially relevant behaviors as primary dependent variables. The findings of the review suggested that future research could use self-monitoring interventions to support an adult with autism in employment settings and that a self-monitoring intervention could be tailored by considering individual differences.

## 1. Introduction

Transitioning into adulthood can be a tremendous challenge for individuals with disabilities. It is particularly challenging for individuals with autism because a successful transition into adulthood requires independence and a multidimensional construct consisting of behaviors known to be challenging for people with autism. Hume et al. [1] argued that independence occurs when “an individual demonstrates the capacity to behave on his or her own” (p. 103). It is important to note that the achievement of independence—for *all* adolescents—is the result of a gradual, multidimensional process. Ideally, during the teen years, an individual’s ability to function independently will increase in more complex settings that lack full support, such as secondary education settings and workplaces. Anthony and Bobzien [2] stressed that as children enter adolescence, they experience physical, emotional, and psychological changes, and this is no less true for adolescents with autism. Regarded as a whole, adolescents with autism are as diverse as any other group. They are characterized by the core set of autism traits, but the range of their individual capacities is great and dynamic. All of these conditions together create a unique learning curve for adolescents with autism striving to attain independence step by step. However, due to their deficits in social skills, communication skills, and executive functioning skills, individuals with autism have a limited capacity for independent performance. For instance, Clarke et al. [3] investigated vocational activity trajectories in young adults with autism and found that continuous efforts to support these individuals were required for them as they worked toward the achievement of independence. Limited independence often leads to low participation in competitive employment, low pay rates, and temporary employment, and it can ultimately lead to unemployment or underemployment for individuals with autism [4].

The prevalence of autism continues to increase. According to the Centers for Disease Control and Prevention [5], an average of one in 44 children is identified with autism. As members of this growing group reach school-leaving age and enter adulthood, interventions that support their learning of functional skills to increase independence and even secure employment opportunities will be desperately needed. Wehman et al. [6] identified one of the critical elements in autism-specific interventions as the use of self-monitoring. Self-monitoring is an important skill not just for adolescents with autism but for all adolescents—a fundamental component of the life stage, essential to the process of growing in independence and reducing any inappropriate or undesirable behaviors. It has been viewed by relevant stakeholders as an important skill for individuals with autism to develop [7,8,9]. Learning to self-monitor affords such individuals the opportunity to take the role of a change agent in their own behaviors [7]. To self-monitor their behaviors, individuals with autism must observe and record the occurrences of a target behavior [10]. One of the benefits of learning self-monitoring is the development of self-determination skills. Self-monitoring is a component of self-management and an initial step for students with autism in achieving self-determination [11]. Individuals with disabilities can learn self-monitoring first and then gradually develop self-management and self-determination skills. 

Individuals with autism may have challenges with executive functioning [1], such as difficulties paying attention or inhibiting inappropriate behaviors. However, research has also documented these individuals’ positive attributes and strengths [12]. Some individuals with autism excel at synthesizing information, paying attention to details, and thinking in pictures. Studies have shown how to tailor self-monitoring interventions that build upon these positive attributes and strengths. For example, Ganz and Sigafoos [13] first surveyed their participants’ preferred rewards. Using these rewards as a goal, participants were motivated to complete a task independently and in a systematic way with visual support. Individualization of the self-monitoring intervention allowed individuals with autism to reduce reliance on others and increase independent task completion.

Several studies have used self-monitoring as part of intervention packages to improve on-task behaviors, most using low-tech self-monitoring interventions. For example, Li et al. [14] developed a low-tech self-monitoring intervention involving self-observation and self-recording, the main material of which was a self-monitoring sheet for participants containing visual prompts and written directions. As technologies improve and change people’s lives, they have been used in education for children and teenagers with autism [15]. For example, Rosenbloom et al. [16] implemented a technology-based self-monitoring intervention to improve on-task and task-completion behaviors. Their results demonstrated that participants were able to complete assigned tasks through the use of a smartphone application, I-Connect, set as a reminder prompting them to complete the tasks. 

The use of technology to build independence and social skills for individuals with autism has been found to be beneficial. Gallardo-Montes et al. [15] described a variety of mobile and tablet applications that can be important resources to support students with autism in developing essential skills. Such applications or software programs target different aspects of learning, such as time management and emotion management. Cañete and Peralta [17] demonstrated how to address the needs of individuals with autism by using assistive technology, beginning with user analysis identifying individuals’ needs and preferences and considering the core symptoms of autism. Many literature reviews have also shown that digital interventions for individuals with autism can help those individuals to develop skills and can facilitate self-monitoring and self-management [18,19,20]. In addition, applications or software programs can be installed on different devices, such as smartphones, iPods, and iPads. However, although the use of technology-based interventions can maximize the accessibility of interventions for individuals with autism, research has yet to conclusively demonstrate the effectiveness of such interventions on the promotion of independence due to the fact that technology-based interventions remain an emerging practice [18].

### Research Questions

Existing and similar reviews have explored effective interventions to help individuals with disabilities to improve independence and reduce inappropriate behaviors. For example, Mize et al. [20] conducted a meta-analysis to explore technology-based self-monitoring systems for students with disabilities and found that such interventions showed a strong effect on increasing on-task behaviors. However, few reviews have focused on technology-based self-monitoring interventions specifically for adolescents and adults with autism. As the population with autism ages, improved ability to work and function independently is one of the most important skills for the achievement of ideal post-school outcomes. The use of self-monitoring to improve independence and/or reduce undesirable behaviors is an imperative need for individuals with autism. 

Thus, the purpose of this research review was to fill a gap in the current literature base by summarizing studies on technology-based self-monitoring interventions for adolescents and adults with autism.The research questions were as follows:What are the participant characteristics in the included studies?What are the characteristics of the technology-based self-monitoring interventions (i.e., delivery formats and types of technology) in the included studies?What are the primary dependent variables in the included studies?


## 2. Method

### 2.1. Article Identification

To identify studies for review, we used a four-step literature search process [21]. First, we searched existing reviews to learn about the current status of research on interventions for individuals with autism [7,8,9,19,20]. The existing reviews helped us to identify the gap we sought to address in this study—namely, the absence of a summary of technology-based self-monitoring interventions for adolescents and adults with autism. We used the existing reviews to identify key search terms. 

Second, we used the following databases to search for appropriate studies: PsycINFO, Education Full Text (EBSCO), Psychology and Behavioral Sciences Collection, and ERIC. We included primary keywords to generate results targeted to “interventions” (e.g., *self-monitoring*) and “participants” (e.g., *individuals or students with autism*); in addition to the primary keywords, we used *technology* as the secondary keyword. Examples of the Boolean search phrases in EBSCO are the following: “(self-monitoring or self-management) AND (autism or autism spectrum disorder)” and “((self-monitoring or self-management) AND (autism or autism spectrum disorder)) AND (technology or computer or tablet or mobile phone or smartphone or internet).”

We used the following inclusion criteria: (a) used technology-based self-monitoring interventions as an independent variable, (b) included adolescence-age or older individuals with autism as participants, (c) was published in a peer-reviewed journal in English during 2010–2022, and (d) used a single-subject research design. According to the World Health Organization [22], adolescence is the phase of life beginning at age 10. Thus, the age of 10 years or older was the inclusion criterion for study participants. Due to the heterogeneity of students with disabilities, it is difficult to find large numbers of study participants with similar disability presentations and similar intervention needs. Therefore, we selected studies with single-subject research designs. The single-case experimental design modality is widely used in the field of special education because it is replicable and tailored to fewer numbers of participants. To examine the direct effects of self-monitoring on a target behavior, we excluded articles that incorporated self-monitoring into a broader intervention package, such as self-management interventions or cognitive behavioral therapies. 

Third, after the initial search, we exported the search results from the databases and imported them into Rayyan (https://www.rayyan.ai, accessed on 13 December 2022), which allows for collaboration and helps researchers to screen articles by choosing the following options: Include, Maybe, and Exclude. For Exclude, researchers can choose a reason for excluding an article, such as wrong drug (intervention), wrong outcome, and wrong population. We also used Prisma to document the search process from identification to inclusion based on the inclusion criteria. To screen articles for the present review, the first and second authors reviewed the titles and abstracts of all included records. Articles were excluded due to irrelevance—for example, wrong interventions, wrong participants, or wrong publication types. After the title and abstract review, the first author reviewed the full text of each included study to identify (a) whether the study incorporated a type of technology to implement a self-monitoring intervention and (b) whether the study participants were adolescents and adults with autism. 

Fourth, for verification purposes, we invited an expert—a faculty member in the Department of Special Education with extensive experience conducting research reviews—to verify the search and coding process. After the four-step search process, 16 studies were included in the present review. Figure 1 presents the Prisma search flow diagram.

### 2.2. Coding 

We used the following categories to code the included studies: (a) participants who met the inclusion criteria (i.e., participants’ age and disability), (b) primary dependent variable, (c) primary intervention, and (d) study design. Specifically, for participants who met the inclusion criteria, we recorded study participants who were individuals with autism aged 10 years or older. If a study had participants who did not meet the criteria, we excluded those participants. The primary dependent variable was the targeted behavior identified by the author(s) of the included study. If more than one dependent variable was targeted in a study, the author(s) would normally identify variables as “primary,” “secondary,” and so on. The present review focused on the primary dependent variable in each included study. We defined the primary intervention as follows: Researchers or interventionists used a self-monitoring strategy, including self-observation and self-recording, to support participants in achieving targeted behaviors and/or reducing inappropriate behaviors. The strategy could incorporate any type of technology. For example, the technology could emit prompts to cue participants or to serve as a platform for them to record their own behaviors (e.g., I-Connect or MotivAider). We recorded the study designs the authors of the included studies reported; it could be any type of single-subject or single-case research design (e.g., ABAB or multiple baselines). We coded the included studies in an Excel document in which we also created a *notes* column to document special features of the studies, such as the self-monitoring training process, different intervention settings, or other special features related to the study design. 

### 2.3. Interrater Reliability

Three coders extracted information from the included studies. The first author is an assistant professor of special education at a top-tier research university. The other two coders were doctoral students in a special education program and a learning, design, and technology program, in their fourth and second years of study, respectively. The first author coded all 16 articles and created a codebook using an Excel document. The coding training procedure had two stages: (a) a review of the inclusion or exclusion of articles and (b) a coding demonstration using the codebook and two included studies as examples. In the first stage, the first author explained the purpose of the literature review and the inclusion and exclusion criteria used to screen and identify studies for the review. In the second stage, the first author demonstrated how to extract information from the included studies and how to use the codes to document the information. The other two coders independently coded seven studies each. All of the coders used the codebook to independently document information from the included studies. The interrater reliability in the present review was collected on 100% of the included studies, and the agreement was 86.25%. The disagreement was resolved by reading the included studies again to discover the correct information.

## 3. Results

A total of 16 single-subject studies examining technology-based self-monitoring interventions for individuals with autism aged 10 years and older during 2010–2022 were identified. Participant characteristics, intervention characteristics, and primary dependent variables are summarized below (see also Table 1). 

### 3.1. Participant Characteristics 

Participants’ genders, ages, and disabilities were documented. A total of 33 participants met the inclusion criteria for this review (i.e., were adolescents or adults with autism). Among these participants, 27 were male, and six were female. Participants ranged in age from 10 to 30 years. Only two studies targeted adult participants. The participant in one of these studies was in a postsecondary education program [30]; the participant in the other study was employed in a medical records position [36]. The participants in the other 14 studies were in elementary or secondary education. All participants had autism; some included studies explicitly described participants as having lower levels of functional intelligence [25,28,32,34] and comorbid disabilities or medical diagnoses, such as attention-deficit/hyperactivity disorder (ADHD) [28,34].

### 3.2. Intervention Characteristics 

For each technology-based self-monitoring intervention, we documented the type of technology used and how the technology helped participants to self-monitor their behaviors. In general, there were three functions that technology performed in self-monitoring. The first was to emit visual, kinesthetic, or aural prompts to remind participants to monitor their own behaviors. For example, a total of seven studies used the I-Connect application to prompt participants [16,24,27,28,31,34,36]. I-Connect is a self-monitoring system that supports users in increasing on-task behaviors. Included studies installed I-Connect on a mobile phone or tablet and the application displayed a visual prompt at a specific time. The visual prompt reminded participants to monitor their behaviors by clicking “Yes” or “No” on the screen. Other studies used kinesthetic prompting. Ganz et al. [29] and Legge et al. [33] used vibrating alarms from a MotivAider device—a personal electronic device that helps people change their own behaviors and habits—to cue participants to record their behaviors on a self-monitoring sheet. Kolbenschlag and Wunderlich [32] used an iPod connected with a single Bluetooth-enabled in-ear headphone to prompt participants with a sound. Participants recorded their on- or off-task behaviors on recording pages at the moment the sound was emitted.

The second function technology performed was to videotape or photograph participants undertaking targeted behaviors. Interventionists used videos or photos of participants themselves to conduct the self-monitoring process [26,37]. Self-modeling photos and videos prompted participants to monitor their own behaviors. Cihak et al. [26] presented self-modeling static pictures using PowerPoint on a handheld computer. One slide presented one photo for participants to watch. Participants recorded their on- or off-task behaviors at the moment the photo was presented. Xin et al. [37] used the Choiceworks application—a learning tool that supports children in completing and managing daily routines—to display participants’ self-images with their recorded voices on an iPad. Participants were able to watch their self-images and listen to their recorded voices to monitor their behaviors. In an intervention similar to self-modeling images or videos, State and Kern [35] videotaped all intervention sessions (i.e., interactive games). After the sessions, the participant watched the videos to determine whether his interactions were appropriate or inappropriate. The videotaped sessions served as feedback for the participant. 

Lastly, technology-enabled interventionists created an electronic checklist with which participants could record their own behaviors. Bouck et al. [25] used an electronic checklist on an iPad for participants to record their food preparation task completion. On the checklist, participants could review recipes and record task steps. Gushanas and Thompson [30] used SurveyMonkey, an online survey tool, to help participants track their daily hygiene. Participants’ self-monitoring consisted of answering a list of hygiene questions on SurveyMonkey. Yakubova and Taber-Doughty [38] used an electronic interactive whiteboard to conduct their self-monitoring intervention. Participants were asked to self-operate on the whiteboard to watch task video clips and to use an electronic checklist to record their task completion. 

### 3.3. Primary Dependent Variables 

The primary dependent variables in each included study were categorized as follows: (a) increasing on-task behaviors or task engagement, (b) increasing skill acquisition, and (c) addressing socially relevant behaviors. A total of nine studies aimed to increase participants’ on-task behaviors using a self-monitoring intervention [16,24,26,27,31,32,33,34,37]. Most on-task behaviors were classroom engagement (e.g., attending to materials or the teacher), engaging in class activities, sitting up at one’s desk, and/or actively completing work during independent tasks (e.g., Clemon et al. [27]). To increase skill acquisition, Bouck et al. [25] targeted food preparation skills for participants using electronic recipes and a self-monitoring checklist. Participants were able to follow multiple steps in a recipe to complete a food preparation task. Yakubova and Taber-Doughty [38] aimed to increase participants’ cleaning skills (i.e., cleaning a mirror, sink, and floor). Participants self-operated an electronic interactive whiteboard to watch task video clips and used an electronic self-monitoring sheet to record their task completions. To address socially relevant behaviors, studies sought to increase appropriate social behaviors or to reduce problem behaviors or undesired habits in a social context [28,29,35,36]. Ganz et al. [29] chose oral self-stimulation and conversation behaviors as the primary dependent variables for two participants. Oral self-stimulation was defined as self-touching tongue, teeth, or any part of the mouth inside and out. Conversation behaviors were asking questions and reducing talking about favorite topics. State and Kern [35] sought to increase appropriate social interactions, such as waiting quietly and at the same time, reducing inappropriate social behaviors, such as interrupting others. Crutchfield et al. [28] aimed to reduce stereotypic behaviors for participants, such as nonfunctional hand gestures and placing hands or objects in one’s mouth. Will et al. [36] targeted inappropriate vocalization behaviors for their participant. These behaviors included aggressive self-talk, swearing, and other inappropriate talking behaviors in the workplace. 

## 4. Discussion

This review identified 16 single-subject studies that used a self-monitoring intervention with technology to improve or decrease target behaviors. This review focused on adolescence-age or older individuals with autism. Most of the participants in included studies were school-aged individuals; in fewer studies, the focus was on adults with autism. Adolescents progressively acquire independence on their path to becoming self-sufficient [1]. Unlike typical peers who may rely on relational supports as they gain independence [39], adolescents with autism can benefit from direct instruction in daily living skills including instruction via video modeling. 

Research stated that adults with autism in employment or postsecondary education struggle to succeed and that intervention to support adults with autism to learn skills is needed [40,41]. Bross et al. [41] argued that some interventions, such as video modeling, can be considered reasonable workplace accommodations in employment settings. Most importantly, the intervention can be naturally embedded in the work routine and supervised by employers and supervisors. Future research could implement such a self-monitoring intervention for adults with autism in postsecondary settings, such as workplaces and community settings. Similarly, adults with autism in a postsecondary program could also use self-monitoring to improve performance. Many postsecondary programs such as Postsecondary Access and Training in Human Services and Aggie ACHIEVE at Texas A&M University are employment-oriented and could incorporate a technology-based self-monitoring intervention to help participants to learn required job skills [30]. 

The summary of studies revealed that different technologies were used to implement self-monitoring, which is in line with the findings of an earlier study [20]. The use of technology in interventions and instruction for learners with autism has been recorded in previous studies (e.g., Odom et al. [19]). The technologies used had different functions across different interventions, but all of these functions served—by means of embedded prompts—to remind participants to monitor and record their own behaviors. The included studies also targeted different primary dependent variables, such as on-task behaviors, skill acquisition, and addressing problem behaviors. This demonstrated that the uses of self-monitoring can be varied, depending on participants’ needs. 

We used *notes* to document special features of self-monitoring interventions. Some studies included participants who had low functional intellectual ability or other disabilities, such as ADHD. Research has suggested that interventions and instructions should consider adolescents’ individual differences, such as their preferred responses and communication methods [42]. Li et al. [14] tailored a self-monitoring intervention by considering participants’ learning needs; they used visual written directions and pictures to help participants with moderate intellectual disabilities learn job-related skills. However, the authors of the studies included in the present review did not clearly report how they tailored a self-monitoring intervention by considering participants’ learning needs or preferences, leaving the question of how a self-monitoring intervention can be used to support different individuals unaddressed. However, the included studies did clearly report how participants were trained to use specific self-monitoring interventions. Some studies used a self-monitoring module to implement the training process (e.g., Huffman et al. [24,31]). Others used videos recorded from baseline sessions (e.g., Clemons, et al. [27]); participants watched self-image videos to distinguish target behaviors and non-target behaviors. Most importantly, studies incorporated goal setting in training [31,36]; participants set goals that motivated them to learn self-monitoring and, ultimately, to achieve the goal. These special features serve as a guide for future research into the design of self-monitoring training for adolescents and adults with autism. 

In addition, since much of the discourse around persons with autism focuses on their skill deficits, it can be easy to overlook these individuals’ strengths. We suggest that future research studies continue to use technology-based self-monitoring and consider participants’ positive attributes, such as paying attention to details and thinking in pictures, to design appropriate interventions. In this way, adolescents and adults with autism will be able to use their strengths to learn important skills to support their employment and/or independent living.”

In conclusion, the present review provides a summary of self-monitoring interventions with technologies to support adolescents and adults with autism. Self-monitoring can be implemented in different settings and can address a variety of target behaviors. More studies are needed to research how to use self-monitoring interventions to support adults with autism. Future studies should also take individual differences into account to reveal how to tailor a self-monitoring intervention based on participants’ learning needs. 

### Limitations 

We acknowledge several limitations in the present review. First, we included only 16 studies in the review. We did not use certain other search methods such as first author search (i.e., searching articles by using the first author’s name to see if there are other similar studies) and ancestral search (i.e., searching articles from reference lists of the included articles) to identify studies. In addition, we did not use other databases, such as SCOPUS, CINAHL, and Web of Science to expand the search scope. Second, we did not include studies that used an intervention package, such as self-management. It is difficult to evaluate the direct effects of self-monitoring on target behaviors if self-monitoring is embedded in an intervention package; thus, we only included studies that clearly reported two components—monitoring and recording—in a self-monitoring intervention. Third, we did not use a set of quality standards to evaluate each included study. A set of quality standards assesses the methodological quality of a study with a single-subject design. Because we did not use a set of quality standards, we were unable to provide information about how rigorous each study was in terms of its use of methodology, although all the studies we included are from peer-reviewed sources. Fourth, since we only focused on the main features of a self-monitoring intervention, we did not document other detailed information, such as interventionist and intervention setting. Based on these limitations, we suggest that future researchers use different search methods and databases to identify studies, use a set of quality standards to assess studies, and record intervention details when summarizing studies. 

## Figures and Tables

**Figure 1 behavsci-13-00138-f001:**
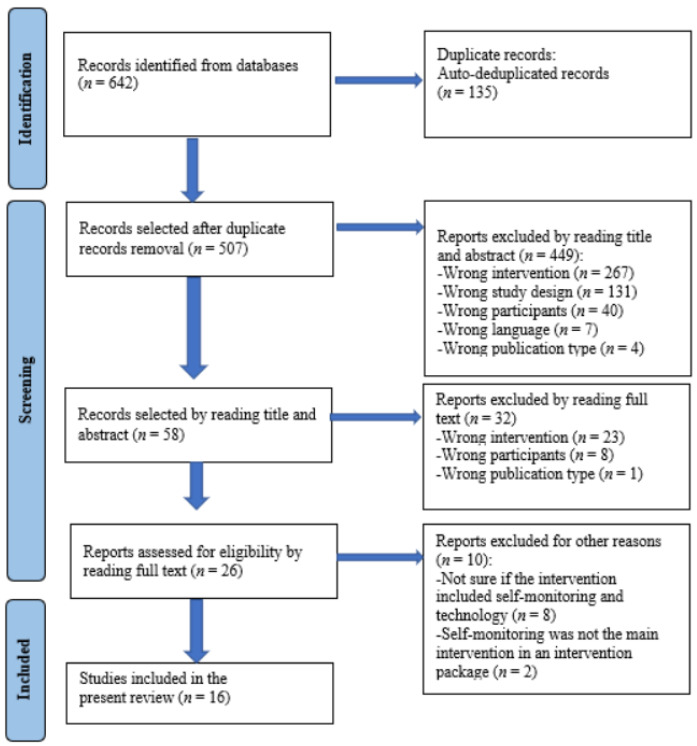
Article identification and screening process (adapted from the PRISMA flow diagram (Page, et al. [23]).

**Table 1 behavsci-13-00138-t001:** Summary of Included Studies.

First Author (Year)	Participant Gender/Age/ Disability or Medical Diagnosis (Other Than Autism)	Primary Dependent Variable	Primary Intervention	Study Design	Notes
Beckman [24]	Participant A: Male/11 years/ Fragile X syndrome. Participant B: Male/10 years.	On-task behaviors.	Self-monitoring with the I-Connect application by responding to the visual prompt at a specific time.	Single-subject ABAB withdrawal.	1. Each student was videotaped from baseline to rate their behavior as on- or off-task for training. 2. Reinforcers were provided.
Bouck [25]	Participant A: Male/15 years/ functioning at the severe level of ID. Participant B: Female/15 years/ functioning at the mild level of ID. Participant C: Female/15 years/functioning at the mild level of ID.	Skill acquisition.	A comparison of high-tech and low-tech self-monitoring: 1. High-tech: An Apple iPad 2 was used to present ingredients, receipts, and the checklist. 2. Low-tech: paper/pencil-based recipes with a self-monitoring checklist.	Single-subject alternating treatment.	The intervention on iPad was more effective than the paper/pencil-based intervention.
Cihak [26]	Participant A: Male/11 years. Participant B: Male/11 years. Participant C: Male/13 years.	On-task behaviors.	Self-monitoring with self-modeling static-picture prompting: 1. Photos of participants self-modeling task engagement were taken. 2. Participants watched the self-modeling photo and monitored their task engagement by marking “Yes” or “No.”	Single-subject ABAB with multiple probes across settings.	All phases occurred in general education settings. Three settings (courses) were targeted for each participant.
Clemons [27]	Participant A: Male/17 years.	On-task behaviors.	Self-monitoring with the I-Connect application by responding to the visual prompt at a specific time.	Single-subject ABAB withdrawal.	1. Videos were used from baseline sessions to provide training. 2. Reinforcers were provided if participants correctly recorded their behaviors.
Crutchfield [28]	Participant A: Male/14 years/Down syndrome/ADHD. Participant B: Male/14 years/ADHD.	Problem behaviors (stereotypic behavior).	Self-monitoring with the I-Connect application by responding to the visual prompt at a specific time.	Single-subject ABAB reversal with multiple baselines across participants.	No additional reinforcers were provided.
Ganz [29]	Participant A: Male/12 years/ speech–language disorder. Participant B: Male/13 years.	Socially relevant behaviors.	Three-phase self-monitoring with a MotivAider device: Phase 1: Participants used the MotivAider only, Phase 2: Participants used the MotivAider and global rating form, and Phase 3: Participants used the MotivAider and tally form to monitor their behaviors.	Not clearly reported.	Reinforcers were provided if participants met the criteria set on the form.
Gushanas [30]	Participant A: Male/22 years.	Distracting body odor.	Hygiene self-monitoring system: 1. Participants were instructed to use SurveyMonkey to track their daily hygiene. 2. Before participants left their dormitories, participants self-monitored their hygiene by answering a list of hygiene questions on SurveyMonkey once per day.	Single-subject multiple baselines across participants.	The researcher recruited observers to observe and collect data for each participant’s level of distracting body odor 7 days a week.
Huffman [31]	Participant A: Male/19 years.	On-task behaviors.	Self-monitoring with the I-Connect application by responding to the visual prompt at a specific time.	Single-subject alternating treatment with two phases of treatment and no treatment.	1. A 20-min self-monitoring module was used to provide training. 2. At the beginning of the training, the participant set an academic goal of achieving an “A” in the course.
Kolbenschlag [32]	Participant A: Male/11 years. Participant B: Male/11 years. Both participants had IQs in the low-average range.	On-task behaviors.	Self-monitoring with a single in-ear headphone connected to an iPod: 1. The in-ear headphone and iPod were used to cue participants to record their behaviors with a sound. 2. Participants used a recording page to record their on-task and off-task behaviors.	Single-subject multiple-baseline across participants.	1. One 20-min session was conducted to train participants to use the self-monitoring procedure. 2. Reinforcers were provided if participants correctly recorded their behaviors. 3. In the maintenance phases, participants could receive a reinforcer for a high level of on-task behaviors.
Legge [33]	Participant A: Male/13 years. Participant B: Male/11 years.	On-task behaviors.	Self-monitoring with a MotivAider device: 1. Participants wore the MotivAider device. The device emitted a prompt to remind participants to record their on-task behaviors. 2. During the intervention phase, the device vibrated at a fixed schedule (every 2 min). 3. During the fading phase, the device vibrated at a variable schedule (every 4–6 min).	Single-case multiple baselines across participants.	Reinforcers were provided if participants correctly presented all on-task behaviors.
Romans [34]	Participant A: Male/17 years/ ADHD/Mild ID. Participant B: Male/15 years/ ADHD.	On-task behaviors.	Self-monitoring with the I-Connect application by responding to the visual prompt at a specific time.	Single-subject ABAB withdrawal.	1. Videos were used from baseline sessions to provide training. 2. Reinforcers were provided if participants correctly recorded their behaviors.
Rosenbloom [16]	Participant A: Male/17 years. Participant B: Male/10 years. Participant C: Male/13 years. Participant D: Male/11 years.	On-task behaviors.	Self-monitoring with the I-Connect application by responding to the visual prompt at a specific time.	Single-subject ABAB withdrawal.	1. A 20-min self-monitoring module and behavioral skills training approaches were used to conduct training. 2. No additional reinforcers were provided.
State [35]	Participant A: Male/14 years.	Socially relevant behaviors.	A comparison of video feedback and in vivo self-monitoring: 1. Video feedback: The participant was asked to watch his behaviors on videotape and respond to the statement “I had appropriate interactions” with a “Yes” or a “No.” 2. In vivo feedback: The participant checked the recording sheet when the prompt on a watch vibrated during the activity session.	Single-subject reversal (ABCBC) across game partners with multiple baselines.	1. Reinforcers were provided if the participant correctly recorded their behaviors. 2. Different partners (teachers and peers) were invited to interact with the participant in the intervention sessions. 3. In vivo self-monitoring is more effective.
Wills [36]	Participant A: Female/30 years.	Problem behaviors (inappropriate vocalizations).	Self-monitoring with the I-Connect application by responding to the visual prompt at a specific time.	Single-subject ABAB withdrawal.	1. The study setting was in the participant’s workplace. 2. Videos were used from baseline sessions to provide training. 3. Goal setting was used.
Xin [37]	Participant A: Female/11 years. Participant B: Female/10 years. Participant C: Female/10 years. Participant D: Male/12 years.	On-task behaviors.	Self-monitoring with the Choiceworks app on an iPad: 1. Choiceworks was used to help participants complete daily routines with images or photos. 2. Participants’ behaviors and voices were videotaped and saved in Choiceworks. 3. Participants watched their self-image of on-task behaviors and listened to their recorded voices to monitor their behaviors.	Single-subject ABAB reversal.	Reinforcers were provided if participants correctly presented all on-task behaviors.
Yakubova [38]	Participant A: Male/16 years. Participant B: Male/19 years.	Skill acquisition and interaction with an electronic interactive Whiteboard.	Self-monitoring with an electronic interactive whiteboard: Participants were asked to self-operate on an electronic interactive whiteboard, watch video clips for each task, and monitor their performance by using an electronic checklist.	Single-subject multiple probes across participants.	1. The daily living skills were cleaning a mirror, sink, and floor. 2. Data were collected through all three tasks in five sessions.

*Note.* We listed the included studies by using the first author’s name in alphabetical order. Based on the inclusion criteria, all participants had an eligibility area for autism. ADHD = attention-deficit/hyperactivity disorder; ID = intellectual disability.

## Data Availability

Data available on request due to privacy restrictions.

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
