# Peer review of "Self-Monitoring Intervention for Adolescents and Adults with Autism: A Research Review"

_behavsci, 2023, doi:10.3390/bs13020138_

Round 1

Reviewer 1 Report

The introduction adopts a novel position about the transition of adolescents with autism to greater independence which is of interest and relevance to the discourse around autism. I would encourage the authors to consider the use of the term ASD in the title and throughout the paper as this is a contested way to refer to people with autism unless it is in the context of a discussion about diagnostic criteria. Using the term ‘autism’ or autistic people is a more widely accepted terminology that is neuro-affirming.  I also encourage the authors to consider framing independence as a gradual process that is not an end point and has gradations for all adolescents, not just those with autism. The current framing implies a standard of independence that is more uniform. A more diverse framing of emerging independence, or increasing independent activities is a more accurate reflection of this life stage. You are also framing the capacity of autistic adolescents as uniformly impaired which is not an accurate portrayal. There is great diversity in the capacity of autistic adolescents and this should be represented in a more balanced way.

Some autistic individuals experience the barriers you describe and may benefit from supports to address this.

The introduction of the concept of self monitoring is helpful and clear to the reader, but would benefit from being contextualised with the importance of this skill for all adolescents. Indeed it is a key part of the adolescent life stage.

You identify difficulties with executive functioning as a barrier for some people with autism in engaging in self monitoring? Are there any strengths that people with autism bring that could be applied to the development of self monitoring skills? Attention to detail, routines, observation skills etc….these would be good to highlight in your introduction.

I also urge you to be  cautious about recognising respectful ways to support young people with autism to learn and to call out coercive or punitive approaches. The token economy approach you refer to in the introduction should be reviewed in this light of maintaining the dignity of the participant.

The final two paragraphs of the introduction bring in the importance of technology as an emerging area of practice with strong examples. This is framed in a positive and respectful way that highlights self determination and agency for young people with autism.

The focus of the review is included in the paragraph entitled research questions but could be highlighted more clearly with a dot point followed by the numbered research questions to help the reader follow along.

You have identified your approach to the search clearly. It would be helpful to explain why you limited the search to the sources identified as there would be other data bases such as SCOPUS and CINAHL and Web of Science that may also be appropriate. You include clearly justified inclusion criteria that have been discussed in the introduction, with the exception of limiting studies to single subject research designs. Including a rationale for this decision would assist the reader.

You need to include a reference/url for Rayyan, to assist the reader with understanding what this is if they are not familiar with it.

The PRISMA diagram is clear and helpful.

You identify that an interrater reliability of 86.25% was achieved on the coding….how did you resolve the differences in coding to arrive at your final coding decisions?

The information provided in table 1 is important and helpful for the reader, however there is a problem with the formatting which makes it unclear which article the information aligns with in the primary intervention column. You need to consider a better way to format this table.

Your summary of the findings from the literature is clear and concise, and is grouped in an appropriate manner.

You do not make an appraisal of the quality of the studies from a methodological point of view which detracts from the quality of the review.

Your discussion addresses the main points identified in the review and answers the research questions. However, as the discussion progresses your language focuses in on students with ASD when the focus of this review was adolescents and adults. It is important to not shift the focus in your discussion from the data that you have identified.

You are thoughtful and open in identifying the limitations of your study. However, I suggest that it would not take too much to do a quality rating of the included studies which would significantly strengthen this study.

This paper provides new and helpful insights into self monitoring supports for young people with autism. I would encourage the authors to frame the discussion in a more functional way, identifying ways to increase agency of the individual with autism and attain the goal of intervention rather than the strong focus on the process of the intervention and the intrusiveness of the self monitoring.

With some rewriting to address the tone and language this paper will be suitable for publication.

Round 2

Reviewer 1 Report

Thank you for your thoughtful and careful responses to the suggested revisions of this manuscript. You have addressed most of the issues I have raised with one exception. 

The point I made about the use of the term ASD or autism spectrum disorder I think was misunderstood. I am suggesting that you should not use the term ASD either in full or as an abbreviation as this is becoming less acceptable to the broader autism community. I suggest that you use the term autism in the title and throughout the paper to reflect contemporary expectations. 
